# Application of Fatigue Damage Evaluation Considering Linear Hydroelastic Effects of Very Large Container Ships Using 1D and 3D Structural Models

**Sang-Ick Lee [1], Seung-Hwan Boo [2] and Beom-Il Kim [3],***

1  System Safety Research Team, Korean Register, Busan 46762, Korea; silee@krs.co.kr
2  Department of Naval Architecture and Ocean Systems Engineering, Korea Marine and Ocean University, Busan 46762, Korea; shboo@kmou.ac.kr
3  Ship and Offshore Technology Team, Korean Register, Busan 46762, Korea
*  Correspondence: bikim@krs.co.kr; Tel.: +82-70-8799-8584; Fax: +82-70-8799-8577

**Abstract:** Owing to the increasing size and speed of ships to ensure economic efficiency, the hydroelastic phenomena of the hull have emerged as an important factor to be considered in the evaluation of strength during the design stage of current ship building procedures. In this study, we established a method to evaluate fatigue strength with linear spring effects using a 1D (one-dimensional) beam model and a 3D (three-dimensional) global Finite Element (FE) model. Firstly, FSI (fluid–structure interaction) analysis was carried out using the 1D beam model of a 15,000 twenty equivalent unit (TEU) container ship. In this step, the method proposed was to calculate the stress RAO (Response Amplitude Operator) of the hot spot points using only the hull girder load from the beam model. Next, a modal superposition analysis was carried out using the 3D global FE model that was directly calibrated to the fatigue damage of the hot spot points. Based on these stress transfer functions with hydroelastic effects, spectral fatigue analysis was carried out, and the portion of linear springing effects in the fatigue damage was analyzed, respectively. These results were compared with the rigid-body-based results in the final design stage. Finally, the applicability of the proposed method at the actual design stage is discussed.

**Keywords:** hydroelastic phenomena; fatigue strength evaluation; linear springing effect; fluid–structure interaction; spectral fatigue analysis

## 1. Introduction

Owing to the increasing size and speed of ships to ensure economic efficiency, the hydroelastic phenomena of the hull have emerged as an important factor to be considered in the evaluation of strength during the design stage of current ship building procedures. Hull structural vibrations, such as springing due to resonance with waves, are more likely to occur in frequency bands corresponding to the high energy of ocean waves. Accordingly, solving problems involving fluid–structure interaction on the hull is becoming increasingly important for hull structural safety. These problems need to be considered during design evaluation. In particular, the need for an efficient analysis and evaluation procedure to be applied to the fatigue strength design of the hull has been gaining attention.

Research on the springing response in ships began with numerical approaches. Bishop et al. [1] performed dynamic calculations using beam theory, such as that proposed by Euler, Timoshenko, and Vlasov, to analyze the behavior of an open thin structure, by comparing the experimental results on uniform, cross-section beams, and it was determined that the Timoshenko and Vlasov beam theory proved to produce relatively efficient results. Lee et al. [2] analyzed the hydroelastic phenomena of vessels using the Euler and Timoshenko beam theory based on the 3D source method. Storhaug et al. [3] conducted systematic research on numerical method comparisons, model experiments, and real ship

measurements. Malenica et al. [4] evaluated fatigue strength by considering the linear springing effect due to wave loads at the hatch corner of large container ships. Kim et al. [5] calculated container ship behavior using the fluid–structural interaction method based on the 1D beam FE model and Rankine panel methods. Senjanović et al. [6,7] and Jung et al. [8] also evaluated fatigue strength by considering the springing effect of ultra-large container ships. Kim and kim. [9,10] and Wang, S. et al. [11,12] developed an evaluation method for ultimate capacity by considering the whipping phenomena of very large container ships. In addition, studies of nonlinear hydroelastic effects such as the second order springing and whipping due to slamming impact were also being studied steadily. Vidic-Perunovic and Jensen [13] conducted a study on nonlinear springing using the second-order strip theory in the wave field, and Shao and Faltisen [14] and Heo and Kashiwagi [15] solved the second-order hydrodynamic force and response using the high-order boundary element method (HOBEM). Adenya, C.A. et al. [16] and Kim and Choung [17] evaluated fatigue damage by considering the nonlinear springing effect of a very large ore carrier. Following this, research for solving the nonlinear hydroelastic problem has been steadily carried out, but it is very impractical to reflect the nonlinear effect in the design stage due to the time-consuming process. Therefore, in order to effectively reflect the analysis results in the design stage, it is most reasonable to calculate the fatigue damage by considering the hydroelastic effect based on spectral fatigue analysis using linear stochastic models, which are currently those most widely used in global shipyards.

In this study, we established a fatigue damage evaluation method based on a spectral analysis approach for linear springing effects on the hatch corner areas of a 15,000-TEU container ship. First, we applied a conventional hydroelastic analysis method to the 1D beam model. When applying this analysis method to a 1D beam model, the stress transfer function cannot be directly calculated at the hot spot points. Therefore, we propose a method to calculate the stress transfer function at the hot spot point using only the hull girder load calculated from the 1D beam. However, this method was applied under the assumption that the upper deck hatch corner, which is a weak section of the container ship, would not be significantly affected by the internal and external pressure. Then, to calculate the fatigue damage at a hot spot location, an in-house program using a 3D FE model-based mode superposition method was applied. Lastly, by comparing and analyzing the results based on the rigid body applied in the final design stage, the characteristics of fatigue damage considering the linear hydroelastic component are described.

## 2. Theoretical Background

### 2.1. Hydrodynamic Model

In our study, the fluid domain was assumed to be a boundary element method, assuming a recently used, three-dimensional potential flow. A Rankine source and a higher-order spline function were used. For the boundary value problem in the fluid domain, a linearized boundary condition at the average position was used. The coordinate system with the ship's forward speed was defined, as shown in Figure 1.

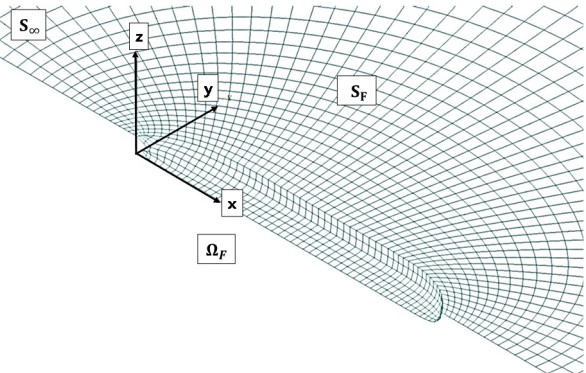

**Figure 1.** Linear pressure distribution.

Here, $\mathbf{\Omega}_F$ is the fluid domain, $S_F$ denotes the free surface, and $S_\infty$ represents the infinite surface. The velocity potential satisfies the Laplace equation, and the solution can be obtained by applying the boundary conditions, where $\phi$ denotes the total velocity potential.

$$\nabla^2 \phi = 0 \text{ in } \mathbf{\Omega}_F \tag{1}$$

The set of the boundary value problems is expressed in Equations (2)–(5). Equation (2) is an expression of the body surface boundary condition. $\mathbf{U}$ represents the ship's forward speed vector, $\mathbf{n}$ is the normal vector on the body surface, $\vec{u}$ is the translation displacement vector, and $S_B$ is the exact body surface.

$$\frac{\partial \phi}{\partial n} = \mathbf{U} \cdot \mathbf{n} + \frac{\partial \vec{u}}{\partial t} \cdot \mathbf{n} \text{ on } S_B \tag{2}$$

Equations (3) and (4) are expressions of the kinematic boundary condition and the dynamic free surface boundary condition in the boundary value problem, respectively. $g$ is gravity and $\zeta$ is total wave elevation.

$$\left[ \frac{d}{dt} + \nabla \phi \cdot \nabla \right] [Z - \zeta(x, y, t)] = 0 \text{ on } Z = \zeta(x, y, t) \tag{3}$$

$$\frac{d\phi}{dt} = g\zeta - \frac{1}{2} \nabla \phi \cdot \nabla \phi \text{ on } Z = \zeta(x, y, t) \tag{4}$$

The open boundary in the free surface is expressed as follows:

$$\nabla \Phi \to 0 \text{ at } S_\infty \tag{5}$$

To linearize the boundary problem, the velocity potential is decomposed into the basis potential ($\mathbf{\Phi}$), incident potential ($\mathbf{\Phi}_I$), and disturbed potential ($\mathbf{\Phi}_d$). Additionally, the free surface elevation is decomposed into the incident wave ($\zeta_I$) and the disturbed wave elevations ($\zeta_d$) (Kim and Kim, 2014).

$$\mathbf{\Phi} = \mathbf{\Phi}(x, y, t) + \mathbf{\Phi}_I(x, y, t) + \mathbf{\Phi}_d(x, y, t) \tag{6}$$

$$\zeta = \zeta_I(x, y, t) + \zeta_d(x, y, t) \tag{7}$$

Through double-body linearization, the free surface boundary conditions of Equations (3) and (4) are expressed as follows:

$$\frac{\partial \zeta_d}{\partial t} - (\mathbf{U} - \nabla \mathbf{\Phi}) \cdot \nabla \zeta_d = \frac{\partial^2 \mathbf{\Phi}}{\partial Z^2} \zeta_d + \frac{\partial \mathbf{\Phi}_d}{\partial z} + (\mathbf{U} - \nabla \mathbf{\Phi}) \cdot \nabla \zeta_I \text{ on } Z = 0 \tag{8}$$

$$\frac{\partial \mathbf{\Phi}_d}{\partial t} - (\mathbf{U} - \nabla \mathbf{\Phi}) \cdot \nabla \mathbf{\Phi}_d = -\frac{\partial \mathbf{\Phi}}{\partial t} - g\zeta_d + \left[ \mathbf{U} \cdot \nabla \mathbf{\Phi} - \frac{1}{2} \nabla \mathbf{\Phi} \cdot \nabla \mathbf{\Phi} \right] + (\mathbf{U} - \nabla \mathbf{\Phi}) \cdot \nabla \mathbf{\Phi}_I \text{ on } Z = 0 \tag{9}$$

The body surface boundary condition is linearized using the Taylor series expansion of the mean body surface.

$$\frac{\partial \mathbf{\Phi}_d}{\partial n} = \left[ \left( \vec{u} \cdot \nabla \right) (\mathbf{U} - \nabla \mathbf{\Phi}) + ((\mathbf{U} - \nabla \mathbf{\Phi}) \cdot \nabla) \vec{u} \right] \cdot \mathbf{n} + \frac{\partial \vec{u}}{\partial t} \cdot \mathbf{n} - \frac{\partial \mathbf{\Phi}_I}{\partial n} \quad on \quad S_B \tag{10}$$

Here, the form body boundary condition is extended to flexible modes using eigenvectors, as follows (Ogilvie and Tuck, 1969).

$$\frac{\partial \mathbf{\Phi}_d}{\partial n} = \sum_{j=1}^{6+n} \left( \frac{\partial \xi_j}{\partial t} n_j + \xi_j m_j \right) - \frac{\partial \mathbf{\Phi}_I}{\partial n} \text{ on } S_B \tag{11}$$

$m_j$ is the nodal mass of the *j*th node and **A** is the eigenvector of the *j*th mode.

$$n_j = \mathbf{A} \cdot \mathbf{n} \tag{12}$$

$$m_j = (\mathbf{n} \cdot \nabla)(\mathbf{A} \cdot (\mathbf{U} - \nabla \mathbf{\Phi})) \tag{13}$$

Here, subscript *j* denotes that the rigid body motions are from modes one to six. The flexible motions are after mode seven. Detailed numerical solutions are given by WISH-FLEX [18].

### 2.2. Fluid–Structure Parameter Exchange

Using a 1D beam FE model, only the displacement at the node along the centerline of the hull can be obtained from the motion equation. The displacement is converted to a displacement of the hull surface, for which the following third-order polynomial shape functions are used.

$$\omega_p = \omega_{n1} S_1(r) + \theta_{n1} S_2(r) + \omega_{e2} S_3(r) + \theta_{n2} S_4(r) \tag{14}$$

$$r = \frac{d}{l}, \quad l = x_{n2} - x_{n1}, \quad d = x_p - x_{n1} \tag{15}$$

$$S_1 = 1 - 3r^2 + 2r^3, \quad S_2 = lr(1 - r)^3 \tag{16}$$

$$S_3 = 3r^2 - 2r^3, \quad S_4 = lr^2(r - 1) \tag{17}$$

Here, $\omega_p$ is the translational displacement in the vertical direction; the translational and rotational displacement at the two nodes are given by $(\omega_n)$ and $(\theta_n)$, respectively, and $S_i$ is the shape function of the beam element. The displacement between the nodes is interpolated by a third-order polynomial in Figure 2. If the displacement at the centerline is converted to displacement at the surface of the object, it is calculated assuming no deformation within the section.

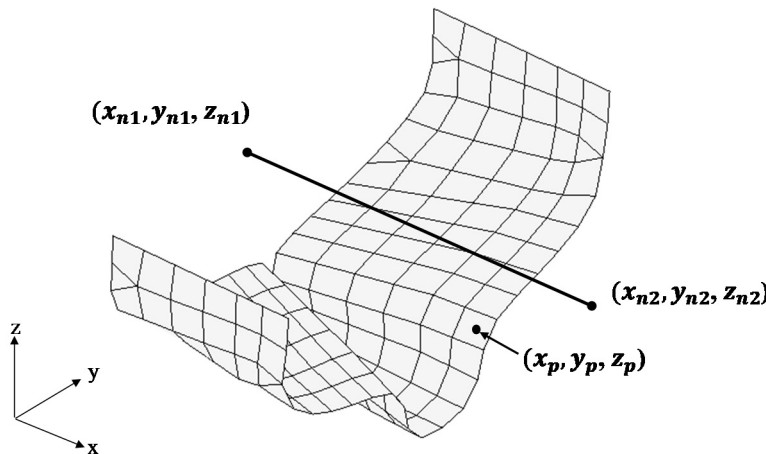

**Figure 2.** Coupling the beam nodes of 1D beam model and grids of 3D panel model.

The beam models are shown in Figure 3, and the hull structures are represented by some 1D beam elements in the centerline of the ship model. The finite element method solves structural reactions, including both rigid and flexible motions. For the calculation of warping distortion, we followed the Vlasoc beam theory, which assumes that the wall behaves like a thin shell. In addition, this beam theory assumes that the contour is not deformed in its own plate and that the in-plane shear strain of the middle surface of the wall is zero. Using a 1D beam model capable of simulating bending torsion and warping distortion coupling, the torsional response of a thin-walled, open-section beam was accurately calculated.

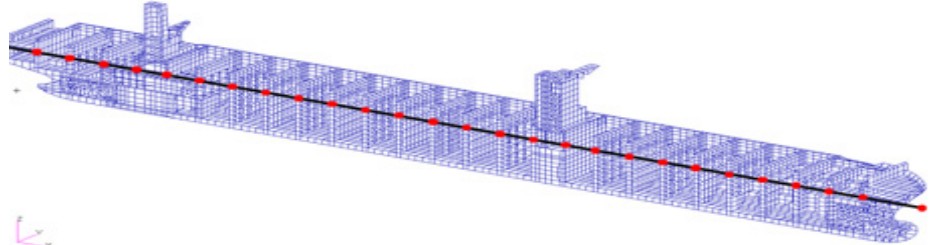

**Figure 3.** An example of a 1D beam model from a 3D global model.

To obtain the displacement of the hull surface from the motion of a 3D FE model, an eigenvector on the hull surface is required. The panel model used in the 3D Rankin panel method does not have the same grid as the structural model. In this study, eigenvectors in a 3D panel model were recalculated through linear interpolation using the location of the grids. Weight functions and eigenvectors can be obtained from the following expressions:

$$[x_p, y_p, z_p] = a[x_{n1}, y_{n1}, z_{n1}] + b[x_{n2}, y_{n2}, z_{n2}] + c[x_{n3}, y_{n3}, z_{n3}] \tag{18}$$

$$\mathbf{A}^j(x_p, y_p, z_p) = a\mathbf{A}^j(x_{n1}, y_{n1}, z_{n1}) + b\mathbf{A}^j(x_{n2}, y_{n2}, z_{n20}) + c\mathbf{A}^j(x_{n3}, y_{n3}, z_{n3}) \tag{19}$$

Here, $\mathbf{A}^j$ is the eigenvector and $a, b, c$ are weight functions. Figures 4 and 5 show how to linearly interpolate the eigenvectors and the correlation of each coordinate value.

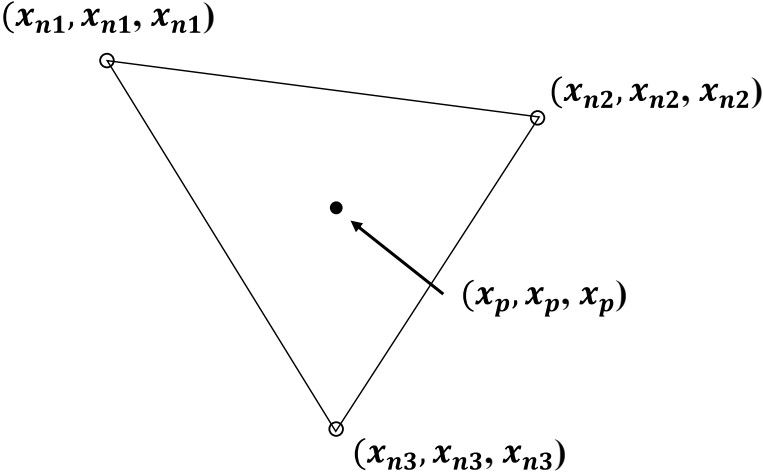

**Figure 4.** Calculation of weight function based on the positions.

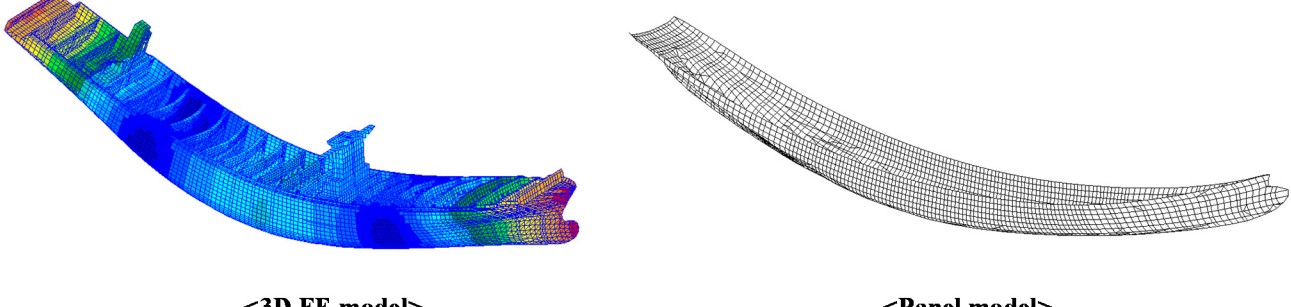

**Figure 5.** Mapping of eigenvectors using linear interpolation.

### 2.3. Stress Calculation from Hull Girder Loads

The global hull girder loads are defined by VBM (vertical bending moment), HBM (horizontal bending moment), and TM (torsional moment). These hull girder moments can be calculated by superposition of the hull girder mode. Under the assumption that the springing vibration response caused by the hull behavior has little effect on changes in local loads (such as external and internal pressure), it was applied to calculate the local stress response using only the hull girder global loads. The target ship was divided into several strips in the x-direction (longitudinal direction), and the loads distributed in the longitudinal direction could be defined as loads on each specific strip. In addition, the vertical and horizontal bending moments could be easily calculated using beam theory. The longitudinal distribution of the torsional load ($M_T$) components acting on each section can be calculated as in Equation (20).

$$M_T(x) = -\int_0^x m(x)dx \tag{20}$$

As the HSF (horizontal shear force) does not act at the center of the shear, the HBM always interacts with the TM simultaneously. In general, it is assumed that the TM acts at the shear center of the transverse section. In this study, it was assumed that the horizontal shear force was applied at a position of approximately 65% of the scantling draught based on the bottom line (in the case of maximum hogging loading in trim and stability, which is generally applied for fatigue strength evaluation). Owing to this assumption, one more constraint introduced by horizontal shear forces was created. The load distribution applied to each node is shown in Figure 6. And is defined as from (a) to (d).

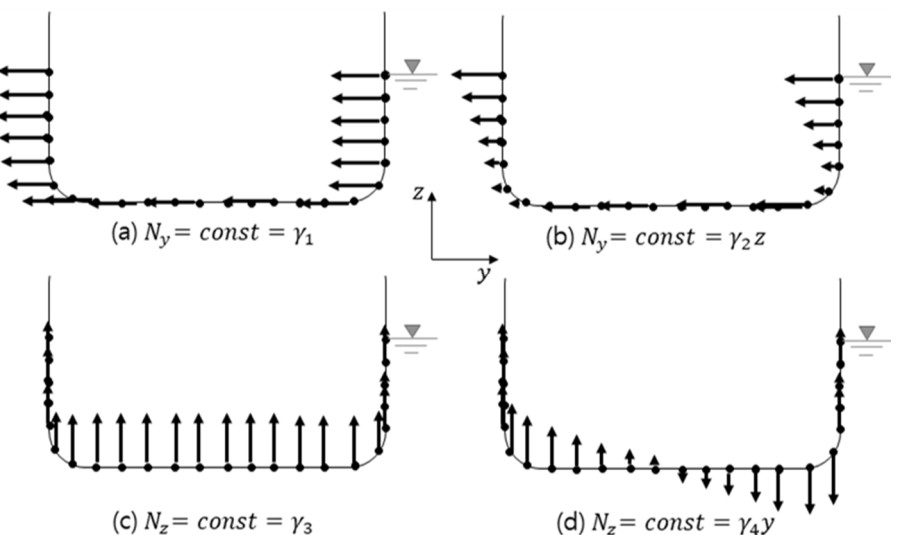

**Figure 6.** Nodal force distribution.

(a) The load in the transverse direction acts uniformly on each node;
(b) The load in the transverse direction changes linearly in the vertical direction;
(c) The load in the vertical direction is applied evenly;
(d) The load in the vertical direction changes linearly in the transverse direction.

Assuming that the number of nodes in the strip section is $n$, the sum of the nodal forces in the transverse direction ($f_H$) is expressed as Equation (21).

$$f_H = \sum_n (\gamma_1 + \gamma_2 y) \tag{21}$$

The moment at the shear center ($SC$) by the previously assumed horizontal force is expressed by Equation (22).

$$f_H(0.65T + SC) = \sum_n (\gamma_1 + \gamma_2 z)(z + SC) \tag{22}$$

Similarly, the sum of the nodal forces in the vertical direction ($f_v$) is expressed as Equation (23).

$$f_v = \sum_n (\gamma_3 + \gamma_4 y) \tag{23}$$

The torsional loads at the shear center ($SC$) by the sum of nodal forces in the vertical direction are expressed by Equation (24).

$$m_T + f_H(0.65T + SC) = \sum_n (\gamma_3 + \gamma_4 y)y \tag{24}$$

Here, $m_T$ is the torsional load generated by the horizontal and vertical loads acting on the nodes of the strip section. Given four equations for the unknown factors ($\gamma_1, \gamma_2, \gamma_3, \gamma_4$), the determinant can be constructed as in Equation (25), and the node load factor can be obtained.

$$\begin{bmatrix} n & \sum\limits_{i=1}^{n} z_i & 0 & 0 \\ \sum\limits_{i=1}^{n} z_i & \sum\limits_{i=1}^{n} z_i^2 + sc\sum\limits_{i=1}^{n} z_i & 0 & 0 \\ 0 & 0 & n & \sum\limits_{i=1}^{n} y \\ 0 & 0 & \sum\limits_{i=1}^{n} y_i & \sum\limits_{i=1}^{n} y_i^2 \end{bmatrix} \begin{bmatrix} \gamma_1 \\ \gamma_2 \\ \gamma_3 \\ \gamma_4 \end{bmatrix} = \begin{bmatrix} f_H \\ f_H(0.65T + SC) \\ f_v \\ m_T + f_H(0.65T + SC) \end{bmatrix} \tag{25}$$

To examine the correlation between the calculated stress and load distribution calculated from the above equations, the following can be done: first, the three hull girder load components (VBM, HBM, and TM) can be uniformly combined in (a) the case of positive direction ($+$), (b) the case of negative direction ($-$), and (c) the case of loads that do not act. Next, by analyzing the stress on the hot spot position through structural analysis, the correlation between the load distribution and the hot spot stress can be analyzed. The stress acting on any position of the hull can be obtained from Equation (26). A more detailed theoretical explanation has been provided by Jung et al. (2020).

$$\sigma_{Total} = \sigma_{VBM} + \sigma_{HBM} + \sigma_{TM} \tag{26}$$

$\sigma_{VBM}$ : The stress caused by VBM;
$\sigma_{HBM}$ : The stress caused by HBM;
$\sigma_{TM}$ : The stress caused by TM.

## 3. Structural Models

The container ship applied for analysis was a 15,000-TEU CLASS. The ship body plan is shown in Figure 7. and details are shown in Table 1.

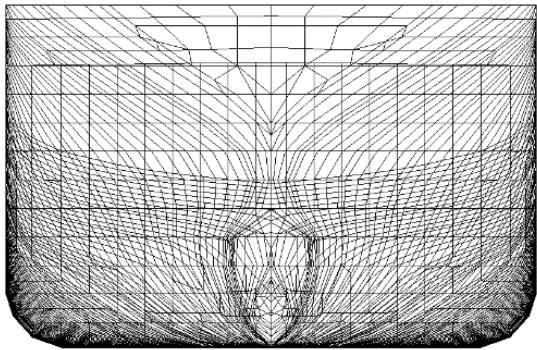

**Figure 7.** Body plan.

**Table 1.** Principle dimension.

| | |
|---|---|
| Length overall, LoA ($m$) | 347.0 |
| Length between perpendiculars, Lpp ($m$) | 342.5 |
| Breadth, B ($m$) | 48.2 |
| Depth, D ($m$) | 29.8 |
| Design draught, Td ($m$) | 14.5 |
| Displacement, ($t$) | 168,658.0 |
| Longitudinal center of gravity, L.C.G (m) | 170.12 |
| Max. speed (knots) | 24.0 |
| Gyration radius in roll and pitch (m) | 17.10, 88.24 |

*3.1. 1D Beam Model*

Vlasov's beam theory is applied to the 1D beam model, which can apply the torsional moment and warping distorsion in consideration of the characteristics of a container ship with a very large opening section. It is a very difficult task to calculate the sectional properties of the hull section, and it is generally calculated through the 2D sectional analysis of the specific cross-section. For the sectional properties of the 1D beam, the grid point was set at the position of the transverse bulkhead in the longitudinal direction, and the information was extracted from the middle of each node. Figure 8 shows the sectional property of each element in the analysis. In order to confirm the dynamic characteristics of the idealized 1D beam model and the 3D wire model, normal mode analysis was performed and the results were compared. As shown in Table 2, it can be seen that the torsion mode (T) and longitudinal bending mode (VB) of the two models are almost similar in dry conditions. This result shows that the idealized 1D beam model reflects the information of the 3D model well.

**Table 2.** Modal analysis.

| | 1D FE Model (Dry) | 3D FE Model (Dry) | Dry Mode Error (%) | 3D FE Model (Wet) |
|---|---|---|---|---|
| VBM (Hz) | 0.604 | 0.614 | 1.8 | 0.471 |
| TM (Hz) | 0.387 | 0.362 | 6.6 | 0.352 |

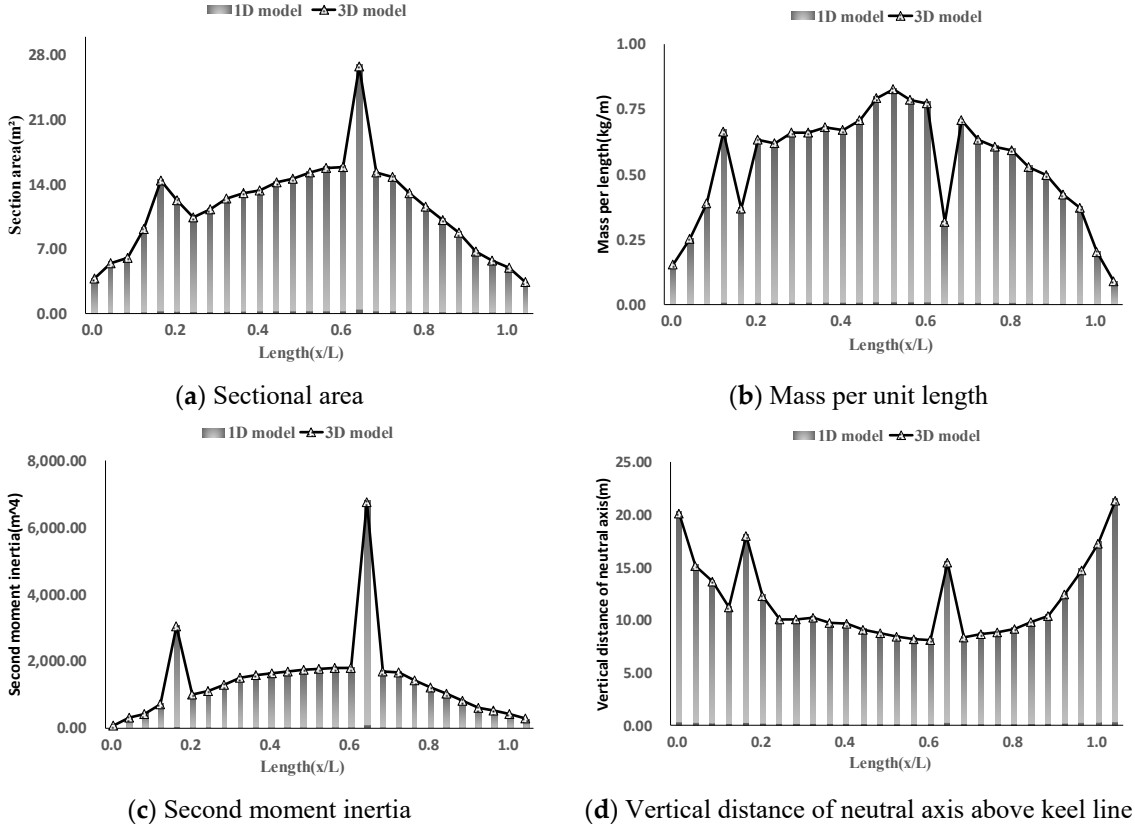

**Figure 8.** Ship property along the ship length.

### 3.2. 3D Global Model

The modal superposition method was applied to calculate the fatigue damage considering the springing response. First, the mode shape and natural frequency of the ship were calculated through mode analysis. Next, for calculating the hot spot stress, the stress corresponding to the mode response was calculated by multiplying this mode response by the stress corresponding to the natural mode response. Where it is not possible to estimate the structural damping coefficients of a ship, values of 2% are generally recommended in the classification guidelines (Kim et al., 2020). Subsequently, the hot spot stress was calculated by superimposing all the stresses corresponding to each mode. The eigenmode analysis results for calculating the structural response are shown in Figure 8. The total number of nodes was 216,000, and the beam and shell element totals were 188,000 and 240,000 in the global FE model. If as many eigenmodes as possible were applied to the analysis, a more accurate stress response could be calculated. However, there were many local modes due to the concentrated mass and boundary conditions of the structural members. Therefore, the stresses at the hot spot points were calculated from six global modes, as shown in Figure 9. For the target ship, two lower torsional modes were identified, followed by a two-node vertical mode.

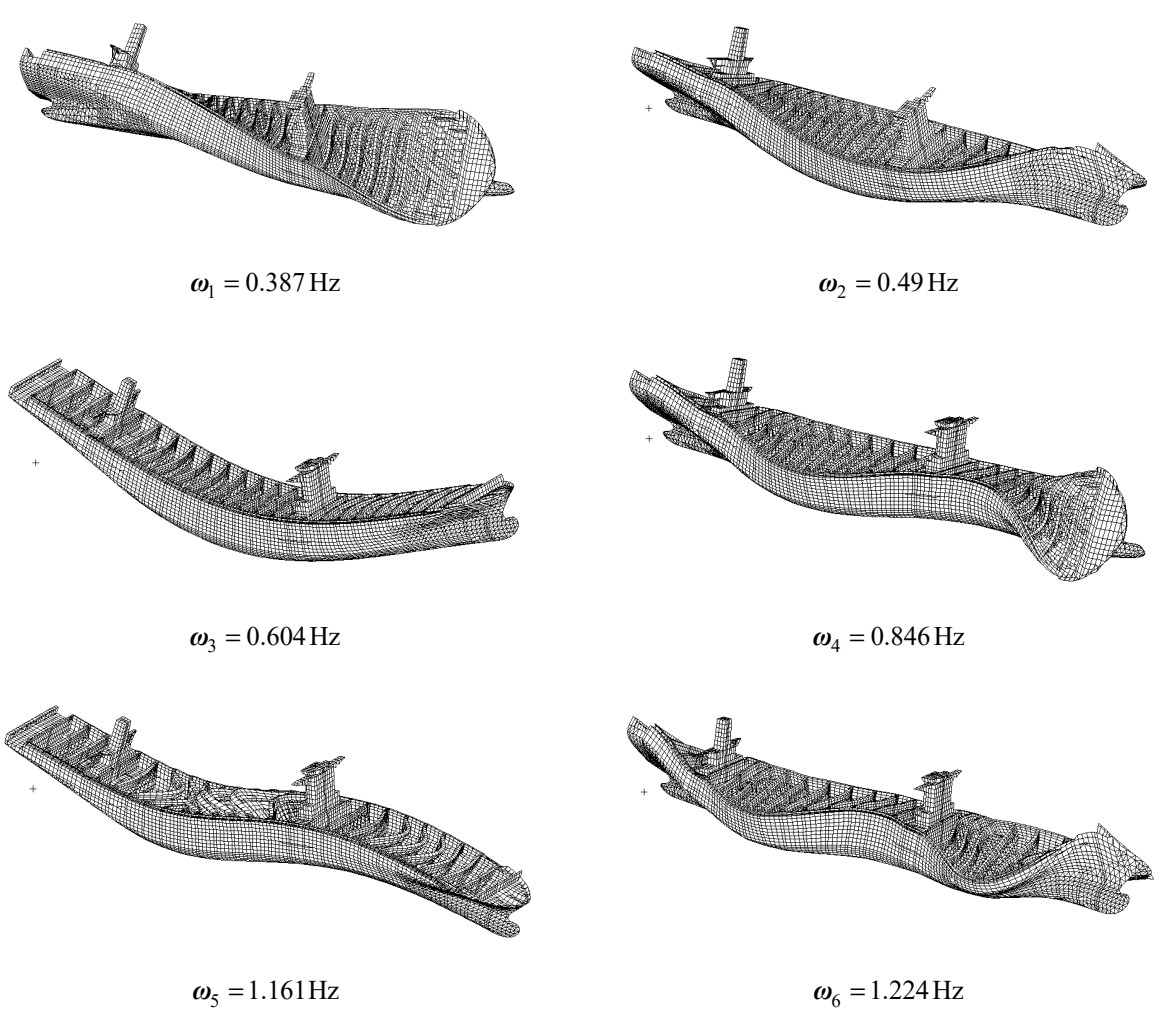

$\omega_1 = 0.387\,\mathrm{Hz}$        $\omega_2 = 0.49\,\mathrm{Hz}$

$\omega_3 = 0.604\,\mathrm{Hz}$        $\omega_4 = 0.846\,\mathrm{Hz}$

$\omega_5 = 1.161\,\mathrm{Hz}$        $\omega_6 = 1.224\,\mathrm{Hz}$

**Figure 9.** Mode shape and frequency.

## 4. Analysis Results

### 4.1. Load Transfer Functions

The hull girder responses for the rigid and hydroelastic models are described. The rigid body motion was calculated by applying only low-order modes (from first mode to sixth mode) of the eigenvector in Equation (13) of Section 2.1, and flexible body motion was calculated by applying six higher-order modes (from 7th mode to 13th mode). The load transfer functions of the vertical bending moments at midship for heading angles of 120° and 180° are shown in Figure 10. Due to wave encounter, it can be seen that the springing response occurred at the lowest frequency range at the head sea wave among all incident waves. Due to the dynamic effect, it can be seen that the response of the flexible body model was greater than that of the rigid body model at low frequencies as well as high frequencies. The load transfer functions of the torsional moments at 0.25 L and 0.75 L are shown in Figure 11. The torsional moment was highest at bow-quartering sea (120°), and the springing response occurred at approximately 1.3 rad/s, as in the VBM response in Figure 10. The horizontal bending moments at 0.5 L are shown in Figure 12. Because the beam sea (90°) was not affected by the wave encounter, the springing response occurred at 2.95 rad/s (=0.471 Hz, the natural frequency of horizontal bending mode in Figure 8).

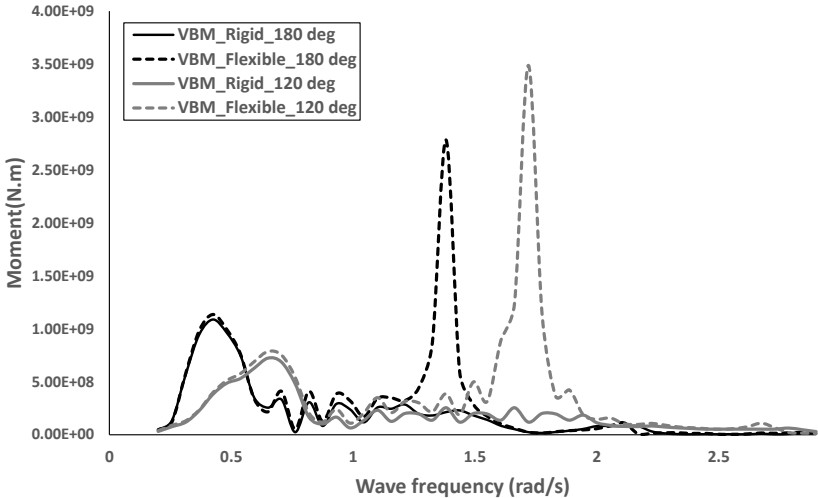

**Figure 10.** Load transfer function VBM (vertical bending moment).

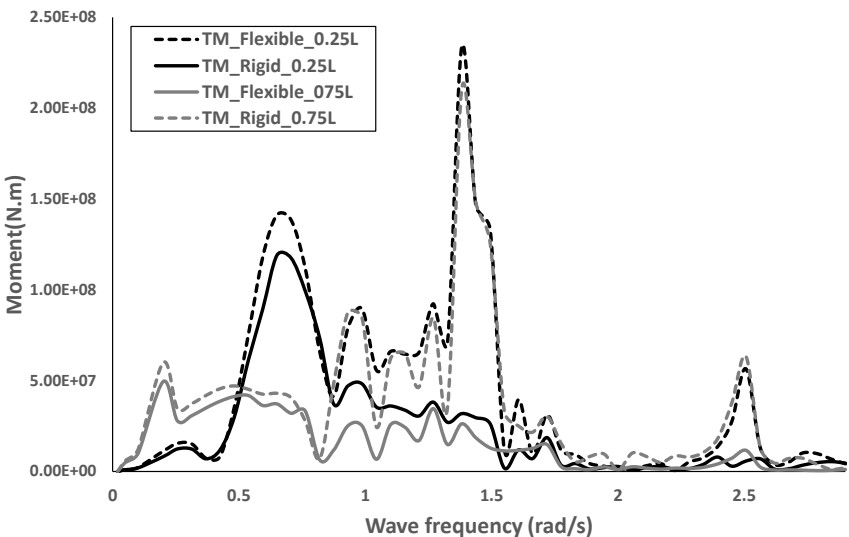

**Figure 11.** Load transfer function TM (torsional moment).

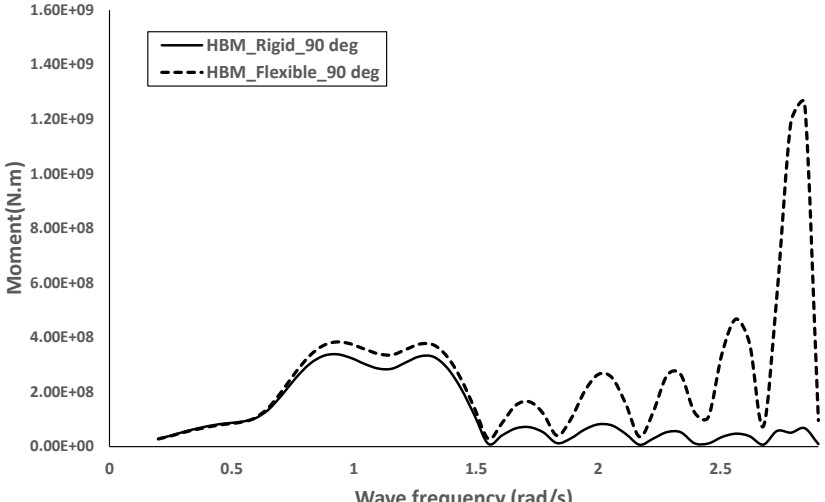

**Figure 12.** Load transfer function HBM (horizontal bending moment).

### 4.2. Stress Transfer Function

Figure 13 shows the hot spot position for the fatigue damage evaluation. To evaluate a hot spot position, such as the hatch corner along the length of the hull (which is a typical fatigue damage evaluation position in a container ship), the intersection of the longitudinal bulkhead and the transverse bulkhead was divided. The stress of the free edge at the hatch corner was used to calculate the fatigue damage.

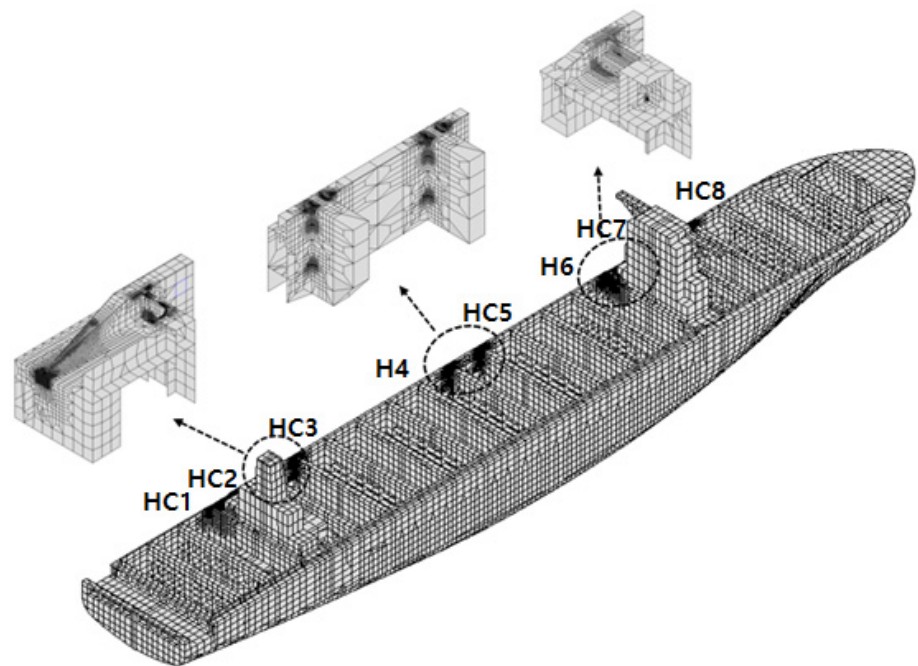

**Figure 13.** Hot spot points for fatigue analysis.

Figures 14 and 15 show the results of the head sea, the bow-quartering sea, and the $L/4$ and near $3L/4$ locations, which are weak points of fatigue strength torsional moments. Here, $L$ is the ship length, and the applied ship speed is 15 knots. As the expression of load transfer function, the stress transfer functions are also expressed as the rigid body component and the flexible body component. There are three stress transfer functions for a bow-quartering sea (heading: 120°) and a head sea (heading: 180°). The meaning of each stress transfer function is as follows.

(a) Rigid body: The stress response calculated by applying only low-order modes (from first mode to sixth mode) of the eigenvector (currently used in the final design stage);

(b) 1D beam model: The stress response calculated by applying the theory of stress calculation from Section 2.3 using the hull girder loads (fluid–structure interaction analysis was carried out and used to calculate the stress response using hull girder moments such as VBM, HBM, and TM);

(c) 3D global model: The stress response calculated by applying the theory of hydroelastic analysis from Section 2.2 (fluid–structure interaction analysis was carried out and used to directly calculate the stress response at the hot spots).

First, in view of the similarity of the responses of (b) and (c), it can be seen that the fatigue strength of the hatch corner of the container can be evaluated using only the hull girder loads. The responses of (b) and (c) regarding the springing effect tended to be slightly larger than or similar to those of (a), with only rigid body responses. In addition, the springing phenomenon occurred at a lower frequency in the head sea than in the bow sea, owing to the effect of the wave encounter frequency based on the ship's speed. As the hatch corner bracket exhibited a strong relationship with the torsional moment, both the

low-frequency hull motion response by wave loading, and the high-frequency springing response by resonance, were greater in the bow-quartering sea than in the head sea.

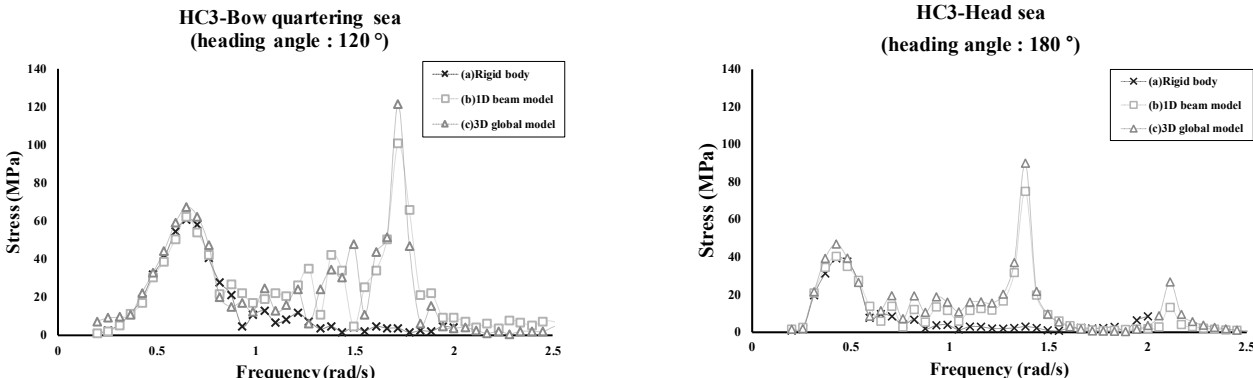

**Figure 14.** Stress transfer function at HC3.

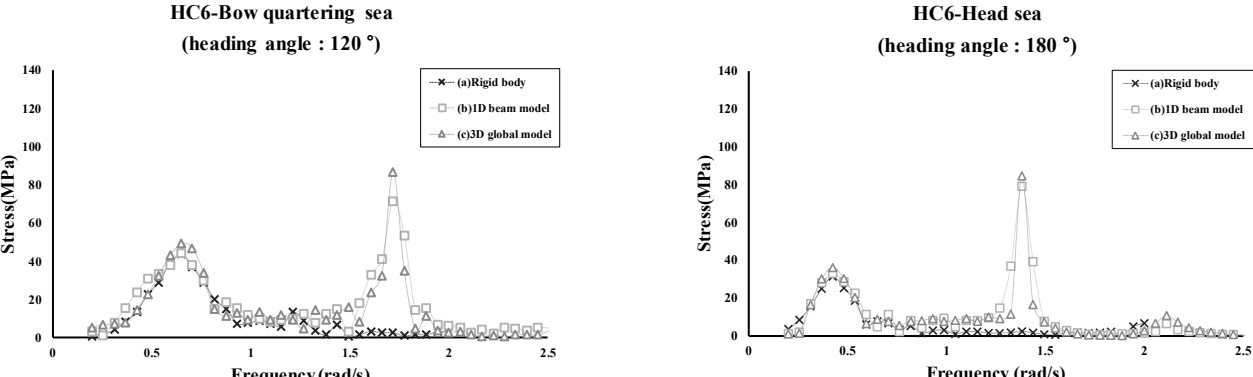

**Figure 15.** Stress transfer function at HC6.

### 4.3. Fatigue Damage

Using the stress transfer function calculated from the rigid and flexible body bases previously calculated, the fatigue damage at the 20 largest hot spot points among the fatigue damages calculated at the hot spot points is shown in Figure 13. When considering the linear springing effect at the point where the fatigue damage based on the rigid body was greatest, it could be observed that the fatigue damage was greater than 30%. This is where torsional moment occurred the most, close to 0.25 L. It can be estimated that the torsional moment increased with the hydroelastic effect and increased the fatigue damage due to the increase in the horizontal moment, which always interacts with the torsional moment.

From Figure 16, when evaluating the fatigue strength evaluation of the hatch corner, which is the weak point of the fatigue strength of the container ship, it can be seen that the results using the 1D beam model and 3D full model were somewhat similar. These results are shown that the stress transfer function of the hot spot location could be properly calculated from the hull girder moment by applying the theory in Section 2.3.

Furthermore, similar final fatigue damage can be obtained even if fluid–structure interaction analysis is performed using other structural models, such as a 1D beam or 3D global model. In other words, this means that the design effect is similar even if a completely different method is used. Moreover, because the structural members in the upper deck-top, such as the hatch corners, do not have internal and external loads, it is confirmed that the initial assumption that fatigue damage can be estimated with only hull girder loads is valid. However, it is considered that additional methodology is required to

apply structural members such as the inner structural member in the ballast tank, where the internal and external pressures directly affect the fatigue damage.

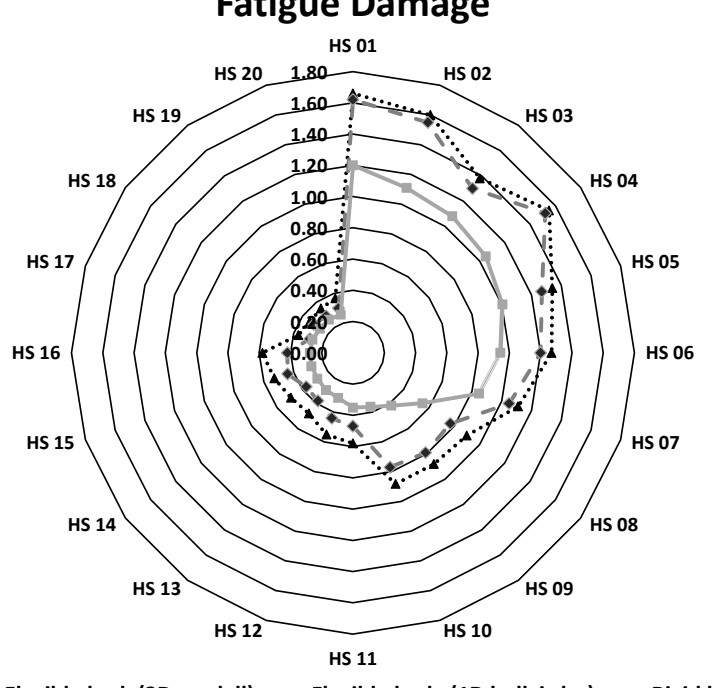

**Figure 16.** Fatigue damage at weak points.

## 5. Conclusions

In this study, we established a procedure for fatigue damage evaluation by considering the springing response of a large container ship based on a widely used linear statistical analysis method using 1D beam and 3D global models. The purpose of our study was to apply the established procedure to a real ship to perform fatigue strength evaluation and to analyze the fatigue damage calculated from the rigid and flexible body theories. Based on the study results described so far, the following conclusions are derived.

- The fatigue damage of the upper deck hatch corner where the internal and external loads do not act can be estimated using only the hull girder loads (VBM, HBM, TM) calculated from the 1D beam model.
- In particular, the location of the hot spot where the fatigue damage is the greatest in a very large container ship has almost the same stress transfer function calculated from the 1D beam and the 3D global model.
- When estimating the fatigue damage by considering the linear springing component, the fatigue damage is increased by 30–100% compared to the rigid body response-based value used in the design stage, and these are caused by the response increase at low frequencies due to waves, as well as by high-frequency responses, such as the springing response.
- In the case of calculating only hull girder loads based on the hydroelastic model at the actual design stage, the fatigue damage at hot spot locations can be estimated using the proposed method.

The fatigue damage results calculated from each application method were analyzed, and their applicability and efficiency were examined at the design stage of the actual ship. Finally, because of the nature of container ships with large nonlinearities, we believe that a study considering the nonlinear components as mentioned in the Introduction is necessary.

**Author Contributions:** S.-I.L. conceptualized and supervised; S.-H.B. analyzed and validated; B.-I.K. wrote the manuscript and analyzed. All authors have read and agreed to the published version of the manuscript.

**Funding:** This research was a part of the project titled, "Development of safety standards for hydrogen fueled ships bunkering and loading/unloading of Hydrogen Carriers (Project NO. 20200478)," funded by the Ministry of Oceans and Fisheries, Korea.

**Institutional Review Board Statement:** Not applicable.

**Informed Consent Statement:** Not applicable.

**Data Availability Statement:** Not applicable.

**Conflicts of Interest:** The authors declare no conflict of interest.

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
