# Peer review of "Application of Fatigue Damage Evaluation Considering Linear Hydroelastic Effects of Very Large Container Ships Using 1D and 3D Structural Models"

_applsci, doi:10.3390/app11073001_

Round 1

Reviewer 1 Report

please download my report in pdf.

Author Response

Thank you for your kind and meaningful comments for publishing the good paper. Please see the attachment.

Reviewer 2 Report

Dear Authors, The paper presents a FE analysis performed on a container ship hull. The authors projected the FSI findings performed on a beam model to a 3D model and performed a fatigue analysis. I find the paper clear and well organized; flowing reading. However, the applied procedure seems to derive from a well established method in the field, so I ask the authors to better underline the innovative contribution. Some minors are listed below: 3D global model: a detailed description of the FE is required (nodes?, elements? llinear elements? etc...) lines 261-275: text is beside the figure 7 Figs 9-11 rigid vs flex body: flex body catches gratest moments between 1-2 Hz, please explain Fig 15 Fatigue damage evaluation should be explained. line 410: 10 is not a reference.

Author Response

(The authors gave the same response as above.)

Round 2

Reviewer 1 Report

all comments have been addressed in the revision.